# Flipping Veterinary Biochemistry, Anatomy, and Physiology: Students’ Engagement and Perception

**DOI:** 10.3390/vetsci11080354

**Published:** 2024-08-05

**Authors:** Christelle de Brito, José Terrado

**Affiliations:** Department of Animal Medicine and Surgery, Facultad de Veterinaria, Universidad CEU Cardenal Herrera, CEU Universities, C/Tirant lo Blanch, 7, Alfara del Patriarca, 46115 Valencia, Spain; christelle.debrito@uchceu.es

**Keywords:** flipped classroom, anatomy, biochemistry, physiology, veterinary, teaching, video, blended learning, higher education

## Abstract

**Simple Summary:**

The flipped classroom (FC) method allows students to first familiarize themselves with educational material independently, and then apply and deepen their understanding through active learning activities during subsequent sessions. Although this approach has been extensively applied in various fields, little is known about its implementation in basic veterinary subjects. This study explores the implementation of FC in veterinary biochemistry; physiology; and anatomy, with pre-class material primarily provided in video format and in-class sessions dedicated to quizzes and interactive activities. The findings reveal robust student engagement. A survey conducted after the first semester revealed that students generally perceived the pre-class material and quizzes favorably, with a significant majority supporting the FC approach. However, by the second semester, preferences evolved, as more students favored traditional lectures over FC, bringing up issues about the pros and cons of FC implementation. Feedback on FC highlighted enhanced comprehension and self-management as key benefits while also highlighting the challenge of time constraints. This study indicates that the adoption of the FC approach can be positively embraced in basic veterinary subjects if it avoids an excessive workload on students. In conclusion, this study suggests that FC can be successfully integrated into basic veterinary education, maintaining a balance that prevents overwhelming students with excessive workloads.

**Abstract:**

Flipped classroom (FC) is a teaching method where traditional learning roles are inverted. Students are provided with material in advance and are expected to study the content prior to in-class sessions. These sessions are subsequently utilized to clarify doubts and examine in greater depth the previously acquired knowledge. Despite the widespread nature of its approach in health education, its application in basic veterinary subjects remains poorly described. This study explores the implementation of the FC approach in veterinary physiology, biochemistry, anatomy, and embryology. Pre-class material was mainly provided in video format, and class sessions facilitated quizzes and interactive activities aimed to reinforce understanding. The findings indicate a high level of student involvement and effective class preparation, as evidenced by over 84% of students participating in FC in-class sessions and generally achieving satisfactory scores on quizzes. A survey conducted at the end of the first semester shows that a high proportion of students positively valued pre-class material (>90%), quizzes (82%), and the FC approach (66%). However, by the end of the second semester, traditional lectures were preferred by more students than FC (45% and 25%, respectively), while 30% of the students mentioned having no preference between the two methods. Analysis of open-ended responses underscored positive facets of the FC approach, including self-organization, enhanced understanding, and availability of pre-class material. However, it also emphasized challenges associated with FC, such as the significant time and effort required. In conclusion, this study suggests that the FC approach can be well received in integrated basic veterinary subjects if it does not imply an excessive student workload, underscoring the potential benefits of a blended teaching approach that combines elements of both traditional and FC methods.

## 1. Introduction

Flipped classroom (FC) is a teaching approach that reverses the traditional learning process: students are provided with material by the instructor in advance and are required to study the content before the in-class session. Pre-class material can be as diverse as guided readings, lecture videos, presentation slides, and practical problems to ensure that students learn the concepts on their own. During in-class time, students have the opportunity to clarify doubts, apply their knowledge, and gain a deeper understanding of the assigned content with the guidance of the instructor. This approach aims to place the student at the center of learning and encourages active engagement [1,2,3,4,5].

Numerous studies and several recent meta-analyses have shown its effectiveness in terms of student preferences and learning performances, as evidenced by its positive impact on academic outcomes [6,7]. However, despite the overall positive trend in the data analysis of FC impact, other studies report the opposite, showing no differences with traditional classes or even negative outcomes [8]. These discrepancies may rely on the high diversity of FC designs and, therefore, further research is needed to improve the method.

The application of the FC model has intensified in the past few years in higher education [9] and particularly in health sciences. In this regard, a recent review indicates that this area has witnessed the greatest number of published studies on FC, surpassing other fields including social sciences, humanities, and natural sciences [1].

Similar to the broader field of health sciences, FC prevalence in veterinary education has also increased. In this respect, FC has found applications in diverse areas, including training students for cytological sampling [10], diagnostic imaging [11,12], introductory animal ethics [13], preclinical science in animal health [14], practical clinical skills development [15,16], and equine nutrition [17]. Furthermore, it has been used to assess cross-cutting competencies like communication in advanced-level students [18] and within professional skills courses for postgraduate veterinary students [19].

In any case, there is a predicted rise in the use of this methodology in veterinary education [20,21]. In this regard, a recent international survey of clinical skills educators showed that out of 101 survey participants representing 22 countries, 42 were already utilizing FC techniques for teaching in a clinical skills laboratory, and 55 others expressed interest in considering the technique for future use in this context [22]. Similarly, a recent international study across universities in the USA, the United Kingdom, and Australia revealed that almost all instructors participating in the survey were familiar with FC, but its application was more limited [23]. Therefore, further exploration of the advantages and disadvantages offered by this type of teaching in various areas of veterinary medicine can provide new insights to enhance its implementation.

The use of FC in fundamental subjects of veterinary education can be particularly interesting for several reasons. FC application enables efficient use of in-class time, which is particularly valuable in basic sciences, a field often grappling with reduced curriculum time [24]. In biochemistry and physiology, where students need to grasp complex concepts and mechanisms, videos can significantly enhance understanding. These resources can be visualized in the classroom or at home and, by adopting the FC approach, valuable classroom time can be repurposed for engaging students in practical applications and problem-solving activities. Similarly, anatomy lends itself well to an instructional method based on visual materials, which can be initially explored at home and then revisited in class for more in-depth understanding. Therefore, the FC method could enhance the understanding of fundamental subjects and promote active learning in basic veterinary education, creating a more dynamic and interactive learning environment where students could explore concepts more deeply.

Multiple applications of the FC have been described across various disciplines within health science, including biochemistry [25,26], physiology [27,28,29], and gross anatomy and embryology [30,31,32,33]. However, its implementation in these disciplines in veterinary education has been sparsely documented. Indeed, to our knowledge, there are no published works reporting the application of FC in veterinary biochemistry or physiology, and only two recent studies, one by De Melo et al. [34] and another by our group [35], have described the use of this approach in veterinary anatomy.

In this study, we present the application of FC in a monocentric study for theoretical classes in veterinary biochemistry, physiology, anatomy, and embryology, all integrated into a content module named “Structure and Function”. We analyze student participation and their perception of the experience. Our findings suggest that, while FC applied to basic veterinary subjects can be well received by the students, the additional workload necessary for class preparation challenges its extensive practical applicability.

## 2. Materials and Methods

### 2.1. Structure and Function Modules in the Veterinary Medicine Curriculum

At the CEU Cardenal Herrera University (Valencia, Spain) veterinary education is organized into modules that integrate several subjects. The modules “Structure and Function (SF) 1” and “SF2” are mandatory and multidisciplinary. They are incorporated into the first year of the Veterinary Degree: SF1 is taught in the first semester, while SF2 is delivered in the second semester. They cover biochemistry, anatomy, embryology, histology, physiology, and immunology. This study focuses on topics included in two SF1 subjects: physiology (cellular excitability) and anatomy-embryology (locomotor apparatus and general embryology), and two SF2 subjects: biochemistry (amino acids and proteins) and anatomy-embryology (organ systems). SF1 and SF2 correspond, respectively, to a total of 12 European Credit Transfer System (ECTS) and 18 ECTS, which represent 120 and 180 training hours divided into theoretical, seminar, workshop, and practical sessions.

### 2.2. Participants

At the CEU Cardenal Herrera University, in the first two years, veterinary teaching is organized into three language groups: English, Spanish, and French. This study involved 70 first-year students who were enrolled in 2022–2023 in the French teaching group in SF1 (*n* = 69) and/or SF2 (*n* = 68). A total of 67 students were enrolled in SF1 and SF2, 2 students were enrolled only in SF1, and 1 student was enrolled only in SF2.

### 2.3. Procedure

This study was approved by the Vice-Chancellorship for Academic Organization and Teaching Staff of the CEU Cardenal Herrera University (Ref PI37A-VV-22). This project was submitted to the Ethics Committee for Biomedical Research of CEU Cardenal Herrera University, which, upon review, determined that, given the nature of this study, evaluation by the aforementioned committee was not necessary (Ref CEEI23 414). Survey participation was voluntary and anonymous. All study methods meet the requirements of the Declaration of Helsinki.

#### 2.3.1. Design of the FC

The students involved in this study had no previous experience in veterinary FC. Therefore, FC was implemented in a limited and gradual manner in theoretical sessions, covering 10% of the theoretical content of SF1 physiology, 15% of SF1 embryology-anatomy, 20% of SF2 biochemistry, and 40% of SF2 embryology-anatomy. The remaining theoretical content was delivered through traditional lectures. Both the FC and the traditional learning were conducted in the same classrooms. Consequently, this study enables an examination of students’ perceptions regarding both the FC approach and traditional lectures. FC was implemented in the first chapter of SF1 physiology and SF2 biochemistry during the initial weeks of the first and the second semester, respectively, and FC for SF1 and SF2 embryology-anatomy were conducted later in the semesters. Two instructors were involved in the preparation of materials, organization, and delivery of classes for both FC and traditional lectures. One instructor handled the anatomy and embryology components in SF1 and SF2, while the other was responsible for the physiology classes in SF1 and biochemistry in SF2. Students were provided with visual material in two different formats. Short videos (mainly shorter than 20 min) were furnished in all the subjects, and annotated radiographs were supplied in anatomy-embryology with identification of the structures that students were required to learn. The teaching videos were created by the instructors (academic teachers trained in video recording and editing), integrating a coherent block of content into each one. The videos were created using PowerPoint slides that were adapted to provide appropriate support for educational videos. When necessary, the videos were further edited using Adobe Premiere Pro. The videos included oral explanations given by the professors, diagrams, photos, animations, and short clips. Keywords and sentences were also integrated to assist students in grasping the central concepts. The content was organized with a clear structure of chapters, and subchapters when needed. Questions were eventually included as a teaching aid to promote student attention and engagement during viewing. Each video was conceived as an independent document and could be viewed independently of the others. Student views or the time spent watching these materials were not recorded.

#### 2.3.2. Pre-Class Material Delivering

During the first week of classes, the students were introduced to the dynamics of the FC, emphasizing the importance of studying pre-class material (generally provided to students at least 5 days before the in-class sessions through the Blackboard educational platform) and informing them that quizzes would be conducted for each FC. The Blackboard platform is familiar and well known by the instructors and the students at the university and easily allows the sharing of documents, folders, links, quizzes, and even large videos between instructors and students. For the preparation of each in-class FC session, students received oral instructions at least once in the classroom, along with a single e-mail. At the beginning of the FC sessions, the students were instructed on the working method, insisting on the fact that previewing the videos and studying them before the classes wasere fundamental for later work in the classroom sessions, which would not be developed as traditional lecture explanations, but as sessions for doubts and in-depth study of certain aspects of the syllabus. To facilitate self-organization, students were informed of the estimated amount of personal working time required to prepare each FC session. This estimation was calculated subjectively by the instructors based on previous experience, the duration of the video(s) or the number of radiographs, as well as the complexity of the content. Instructors paid attention to ensure that the personal work did not exceed one hour for each in-class session. To facilitate effective note-taking during video visualization, students were provided with the corresponding slides from the PowerPoint-based videos.

#### 2.3.3. Topics

FC was applied to various topics in SF1 and SF2 subjects (Table 1).

#### 2.3.4. In-Class Sessions and Quizzes

In-class attendance to theoretical SF1 and SF2 classes is not mandatory. The in-class sessions were generally initiated with students taking an online quiz using either Microsoft Office 365 FORM or the Blackboard educational platform. Afterward, the results of the quiz were reviewed in the classroom, focusing on the questions with the highest percentages of errors. Furthermore, topics related to essential definitions were discussed, along with exercises on theoretical concepts and reflections on practical approaches, and mistakes and doubts were discussed. The class concluded with a discussion focusing on less clear or misunderstood aspects of the pre-class material. Instructors could also use the in-class sessions to review key points from the pre-class material, expand knowledge, and address applied and advanced problems and concrete examples.

The quiz for each in-class FC session consisted of 4 to 10 questions, all questions having the same weight in a given quiz, and the score for each quiz was normalized to 10 points (Table 2). The questions were either of “true or false” type or had 4 possible options with only one correct answer. The questions in these tests were based on the content that had been previously made available to students in the pre-class material, with the aim that they could be answered or deduced from studying the provided material. The students received immediate feedback on their quiz results through the online correction system. Additionally, the proposed questions were discussed in class, with a particular focus on those where the results had been weaker.

#### 2.3.5. Quiz Contribution to SF1 and SF2 Final Grade and Assessment of SF1 and SF2

The completion of the quizzes (not the score obtained by the students) was considered as one item among others in the student participation grade. Participation represents 10% of the total grade of SF1 and 5% of the total grade of SF2. Thus, it is assumed that the completion of the quizzes represents less than 2% of the total grade in both modules.

#### 2.3.6. Initial Control of Student Engagement and Satisfaction with Pre-Class Material

In the first in-class FC session, students completed an anonymous questionnaire using Microsoft Office 365 FORM. This questionnaire was designed to early monitor student engagement and satisfaction with the pre-class material. It included two questions for each video: one “yes or no” question asking whether the student watched the video or not, and a scale measure of 0–10 points to determine if the student would recommend the video to veterinary students who were just starting their education.

The theoretical evaluation (which affects the teaching described in this work) of SF1 and SF2 was carried out through theoretical exams featuring multiple-choice questions and short-answer questions. Students were informed about the types of exams from the beginning of each semester. The exams combined questions that directly assessed the knowledge acquired by the students with others that evaluated deeper learning, connecting concepts. To help students familiarize themselves with the exam format, various practice sessions and different types of tests were conducted, including in-class Kahoot quizzes and self-assessment tests outside the classroom.

### 2.4. Surveys and Data Analysis

To study the students’ satisfaction with the FC teaching system, two different surveys were administered. The first survey was conducted at the conclusion of the first semester (survey 1), and the second was carried out at the end of the second semester (survey 2). The purpose of the surveys was explained to the students, and they were informed that the data would be used for academic and research purposes only. Participants were also informed that answering the questionnaire was voluntary and that responses were collected anonymously. The main objective of the surveys was to determine the overall impact of this methodology on the attitude and learning experience of the students. Surveys were based on a previously published questionnaire [35].

Survey 1. Students were asked about the importance of class preparation, their implication in class preparation, the usefulness of videos and annotated radiographs provided before class, the usefulness of the quizzes, and their perception of the FC approach. For each item, students could give their opinion on a 1 to 5 scale, with 1 meaning “strongly disagree”, “never”, or “not important”, and 5 meaning “totally agree”, “always”, or “very important”.

Survey 2. Students were asked about their preferences between traditional and FC teaching methods in SF, or whether they had no clear preference for either teaching approach.

Students were also given the possibility to express their opinions with open responses.

No demographic data were collected.

The descriptive analysis of the data and the corresponding figures were performed using Excel. In some graphics, a linear trend curve was added. This curve corresponds to the graphical representation of data points that illustrates the general trend of change while assuming a linear relationship.

## 3. Results

### 3.1. Student Engagement in the In-Class FC Sessions

#### 3.1.1. First Semester

The implementation of FC strategies in SF1 commenced as early as the first week of the academic term and continued throughout the semester. They applied to selected topics of veterinary physiology and anatomy-embryology, while the remaining subjects were delivered through traditional lectures.

During the first in-class FC session, a brief questionnaire was conducted as an internal measure to evaluate the initial system’s acceptance among students. This questionnaire aimed to determine whether newly enrolled first-year students at the university were adopting the FC method and if they were satisfied with the provided pre-class material. The questionnaire whether the students had viewed the first two videos provided in physiology and, if affirmative, requested their rating on recommending these videos to first-year veterinary students on a scale from 0 to 10, with 0 signifying “definitely not” and 10 indicating “definitely yes”.

A total of 92% of the students (64 out of a total of 69) answered the questionnaire and almost all of them (more than 95%) reported having watched the videos (63 for the first video and 62 for the second video, out of 64) and rated their recommendation of the videos as 8 or more out of 10 (60 for the first video and all 62 who watched the second video) (Figure 1).

In addition, student involvement in in-class FC sessions was studied. Interestingly, the students’ attendance was consistently high throughout the entire semester, with active participation in the quizzes conducted in each session (Figure 2). Indeed, upon analyzing students’ engagement with quizzes, the findings revealed a nearly total participation rate, with almost all students completing the questionnaires provided, with an overall average participation rate of 92%. The initial quizzes were completed by over 90% of the students. Intriguingly, a slight downward trend in participation was observed during the semester, with the last quiz corresponding to the lowest participation rate of the semester (84%).

Although the scores of the quizzes were not considered for the students’ final grade, the answers were reviewed to detect the most frequent errors and discuss them in class. In this regard, it is worth noting that the quiz scores were quite variable during the semester (Figure 3), with the highest marks observed in in-class FC sessions 1, 2, and 5 (introduction to cell biology, cellular transport, and radiology of the axial region), and the lowest in sessions 3 and 6 (embryonic manipulations and limbs radiology). The analysis of the responses showed that the questions with the lowest percentage of correct answers in those quizzes were those whose answers did not come explicitly from the provided material content but required a higher degree of reflection or interpretation. This, apparently, presented a greater level of difficulty for the students compared to the other quizzes. Additionally, relatively high variability in scores for each quiz, along with a downward trend in grades over time, were observed.

#### 3.1.2. Second Semester

During the second semester, the FC method was implemented in SF2 for selected topics of biochemistry and anatomy-embryology. Like the pattern observed in the first semester, nearly all students engaged in the quizzes, with generally more than 90% of the students taking the quizzes (Figure 4). As described for SF1, a diminishing trend in participation could be observed in SF2 quizzes over time, although it was much less pronounced, with slopes of −2 and −0.8 observed in the first and second semesters, respectively. In a similar way to the first semester, the overall average participation rate for SF2 remained at 92%.

An analysis of the quiz results showed that the scores were more homogeneous in the second semester (Figure 5) as compared to the first semester.

### 3.2. Students’ Satisfaction Survey

A satisfaction survey was provided to the students at the end of the first semester (i.e., SF1), asking for perceptions of different FC aspects. This survey was completed by 88% of students (61 out of 69).

Almost all the students (93%) who responded to the survey admitted agreeing or totally agreeing that working on pre-class material is important for their learning, and the same percentage stated that they worked on the material provided several times or always. When asked about the utility of the pre-class material, students expressed that they considered both the videos and the annotated radiographs as either considerably or very important (90% and 93%, respectively). The usefulness of quizzes was slightly less valued (82%). Finally, students were asked whether FC teaching would be positive in SF1, with 66% agreeing or totally agreeing with the statement (Figure 6).

In the open-ended responses, students shared a range of comments, among which representative comments are included below:-“I appreciated this way [FC] of working because it allows for a more enjoyable class with an exchange of questions that helped me clarify my doubts”.-“I think having the class material before lectures is a plus for understanding. However, I find that having to systematically work on the material before lectures and taking time can be counterproductive if the material is difficult to understand”.-“The educational material provided is of great usefulness for learning”.

Finally, a second survey was conducted to evaluate the satisfaction level of the students with the FC teaching system at the end of the second semester (i.e., for the SF2 subject). This semester, as previously explained, featured a more intensive implementation of FC. The survey directly asked about student’s preferred teaching method (FC, traditional lectures, or both equal). This survey was completed by 78% of the students (53 out of 68). In this case, traditional lectures were favored by 45% of the students, while 25% preferred FC and 30% had no clear preference (Figure 7).

To investigate the preference of students between the two systems, open-ended responses were collected and analyzed. Students’ comments underscored several advantages of the FC, notably self-organization, enhanced focus, improved comprehension, in-class doubt resolution, and the opportunity for unlimited review of pre-class material. However, drawbacks were also highlighted, particularly the demanding time and effort required for FC and the difficulty of integrating pre-class material study into personal schedules. Some students also noted that FC could lead to decreased focus during in-class sessions due to prior material preparation and a perceived lack of engagement compared to traditional lectures. Ultimately, comments suggested that traditional lectures and FC could complement each other as two distinct teaching approaches.

Some representative students’ comments are shown below:-“FC allow us to already have the concepts before class, making it easier to understand them during in-class sessions”.-“FC take up a lot of our free time. I do like FC and being able to work on them at home beforehand, but sometimes it takes up a lot of time during our weekend study time”.-“I prefer traditional classes, but the FC approach allows me to prepare well and understand my lesson”.

The results emphasize that, while some students appreciated the opportunity to prepare and familiarize themselves with the course content at their own pace, they generally perceived the time-consuming FC characteristic as a critical drawback.

## 4. Discussion

This study introduces the implementation of the FC methodology in basic subjects of the first-year veterinary curriculum, specifically targeting theoretical aspects of anatomy and embryology, physiology, and biochemistry. The research aims to assess students’ engagement and perception of the FC approach.

The adoption of FC has witnessed a notable surge in both general education and veterinary medicine in recent years, as evidenced by several studies [1,23,36]. However, to our knowledge, this is the first work that shows its application across different basic subjects such as anatomy, physiology, and biochemistry in an integrated manner within the field of veterinary medicine.

FC is considered to optimize time and increase student engagement in the learning process, typically leading to enhanced educational outcomes. However, its successful implementation relies on the students’ self-discipline, requiring them to effectively organize their pre-session work. Additionally, a well-designed teaching structure is crucial to coordinate and support students’ efforts [1,5,32]. Therefore, gaining insights into how students approach this instructional method is highly interesting to establish a suitable learning framework that addresses their educational needs and ensures effective learning outcomes.

In this study, the most effective elements for FC learning, according to the students, are the materials provided before class. While the evaluation quizzes used at the beginning of the face-to-face sessions were well received, they did not reach the same level of acceptance. Additionally, one of the factors that may influence the acceptance of the FC is that some students consider this type of teaching to be more time consuming.

### 4.1. Importance of the Pre-Class Material and Pre-Class Work

The success of FC depends, to a large extent, on the preparation of students, as it enables them to initiate their learning and contributes to in-class session success. Thus, promoting student engagement in out-of-class learning is a significant consideration when implementing the FC approach [37]. In this sense, a substantial 93% of the students concurred or strongly concurred that prior work is important for their training. Therefore, special attention is required in selecting adapted information supports to enhance student adherence. The current student generation, often referred to as Generation Z, has grown up in a digital era characterized by smartphones, widespread broadband access, and instant access to information. Generation Z shows a preference for learning through video content rather than traditional reading methods. However, the impact of this preference on critical thinking development remains uncertain [38].

In this study, the pre-class material was mostly provided to students in a short video format. To precisely align with the content of FC topics and adequately prepare students for in-class sessions, the instructors participating in this study created new audiovisual resources. Alongside this approach, two notable aspects were observed.

Firstly, even though there was a substantial repurposing of existing teaching slides into the created videos, it led to a significant surge in the instructors’ workload. This is in line with other observations, highlighting the major challenge posed by the creation of FC material [37]. In this context, we would like to emphasize the importance of organizing time management and workload distribution strategies to enhance the effectiveness of FC implementation. One potential goal is to reduce the time and effort required for content creation. Implementing a structured schedule for content creation and distribution can aid instructors in better managing their workload. Ultimately, by prioritizing these strategies in FC implementation, instructors can enrich the learning experience for students and establish a more sustainable and efficient FC environment in veterinary education. Secondly, this customized audiovisual material was well received by the students, consistent with findings from other authors. Its value as material for FC preparation has been demonstrated, for example, in human anatomy classes [39,40,41,42], and veterinary medicine areas such as anatomy practical sessions [35] and animal welfare [43]. Similarly, various studies have shown high student appreciation for videos as FC material in biochemistry [43,44,45] and physiology [46]. Typically, videos are recommended not to exceed 15–20 min, as longer videos may lead to disengagement [38,47]. In our case, the videos adhered to this guideline, with most ranging from 5 to 15 min. However, recent research questions whether this duration can be a general rule. Indeed, a survey of students specializing in physiology revealed marked disparities in their preferences for different durations of instructional content, specifically between a concise 15 min mini-video and an extensive 45 min video, suggesting the importance of individualizing or segmenting material preparation according to the characteristics of different student groups [46].

### 4.2. In-Class Sessions

Student engagement was confirmed by their participation in the in-class sessions, maintaining a high level of involvement throughout the semesters (over 84%). This is noteworthy, especially considering that these sessions are not compulsory and that the quizzes administered at the beginning of the sessions contribute minimally to the final SF1 and SF2 grades. Therefore, this figure suggests a high level of engagement and interest among the students. However, a slight decrease in participation was observed over time in the two semesters. Reduction in student involvement over the span of the semesters has also been described in other studies and is generally considered to be a consequence of the increased workload that usually occurs at the end of the terms [35,39,47,48]. The slight decline observed during the semesters may be attributed to this circumstance. This trend was less apparent in the second semester, perhaps due to better organization among students, who better understood and assimilated the course dynamics.

Importantly, while most students acknowledged the quizzes were beneficial, they considered them less useful than the pre-class material. It is possible that students do not specifically perceive this as part of their preparation since it is not material they actively work on. Quizzes have been described as a key element in increasing student motivation and the effectiveness of FC [6] and we believe they are important tools to assess student readiness for the in-class sessions and guide session activities. In addition, these quizzes make it possible for the instructors to diagnose students’ work, identify misunderstandings of the content, provide early feedback on students’ learning, and detect students who might need additional help to improve their academic level.

In our setting, implementing the FC approach did not change instructional time; however, it enabled a substantive transformation of the pedagogical activities carried out within that timeframe. Indeed, as the students learned the content before coming to class, it allowed more time during in-class sessions to be allocated for different and complementary tasks. First, instructors could recall key points, which is vital for memory retention. Individuals process information for retrieval later. If the information is not used, it is less likely to move into a long-term memory store [49]. Moreover, class time could also be used to discuss and debate knowledge and deepen student understanding, encouraging students to contemplate questions related to essential definitions and vocabulary important for understanding the subject. Additionally, students could engage in exercises centered around theoretical concepts or in interpretations of practical experiments and explore illustrative examples to reinforce the central ideas of the subjects.

### 4.3. General Assessment of the FC by the Students

In this study, students received instruction through two methods: FC and traditional lectures, enabling an evaluation of their perceptions of both systems. In the first survey conducted after the six FC sessions in SF1, a substantial majority of the students expressed a positive perception of FC teaching, as 66% of students agreed that this system could be beneficial for their learning. However, in the second survey following the eight FC sessions in SF2, a greater percentage of students showed a preference for traditional lectures over FC (45% versus 25%, respectively). Although different studies have reported contrasting findings (see [6] for review) about FC efficacy, similar considerations have been described among students in pharmaceutical calculations and algebra who preferred traditional classes [50,51]. While in our study a possible ‘instructor effect’ could be considered a priori, influencing a more favorable perception in one semester compared to the other, this seems unlikely given that the instructors remained consistent across both semesters. Therefore, it is probable that other factors contributed to the decline in FC appreciation. In this regard, it is interesting to examine the open-ended responses provided by students to the questionnaires. While students appreciated the opportunity to structure their work in their own way and to clarify doubts during in-class sessions, they also emphasized the substantial time investment required for FC session preparation. Therefore, workload accumulation or greater intensity in the implementation of FC in the second semester may be an important cause of the decrease in the appreciation of the system.

It is worth noting that the scores of the pre-class quizzes exhibited a non-negligible heterogeneity. This observation suggests that some students adapted better to the FC system than others and may also be a contributing factor to their varying levels of appreciation for FC. Indeed, this type of teaching likely requires better time management and may be better understood among students with higher grades. This hypothesis aligns with the findings of a study conducted in a course on equine nutrition, where high-achieving students tended to hold more favorable views and attitudes towards FC/peer-assisted learning compared to low-achieving counterparts [17].

An important aspect to consider in the development of FC is that this type of teaching, according to students, sometimes requires more dedication time, mainly because the traditional lecture format requires less out-of-class preparation [33,52,53]. The previewing of videos, supplementary work materials, or group discussions that FC often involves can be perceived by students as more time consuming [54,55]. In our case, although instructors took special care to ensure that out-of-class work with provided materials did not result in an excessive increase in dedication time, some students did not perceive it this way in the second semester. This was likely because the number of FC sessions increased compared to the first semester. In this context, combining different teaching approaches may improve the overall educational experience for students. In our study, the combination of time-consuming demands associated with the FC approach and the lack of dedicated time for preparing FC sessions in students’ academic schedules could have contributed to a feeling of work overload among students at certain times. Therefore, it is essential to consider FC application in conjunction with other student activities scheduled for specific dates to avoid work overload. These observations also suggest that incorporating FC can be effective in basic veterinary subjects by achieving a balance that prevents overloading students with excessive workloads. Thus, it might be appropriate to mix the FC approach with other more traditional teaching methods. In this sense, analysis of open-ended responses revealed that students themselves suggested that the implementation of a blended learning system seemed an appropriate strategy.

In this regard, advancements in digital technologies such as 3D printing have facilitated access to specific specimens, while virtual and augmented reality have been successfully applied, particularly in veterinary anatomy [56,57]. These innovative teaching methods cannot serve as the sole means of knowledge dissemination and combining these methods with FC offers an interesting approach to ensuring an engaging and efficient teaching and learning process, that could favor deeper learning.

Additionally, the introduction of game-like and interactive elements, such as challenges and polls, into both pre-class material and in-class sessions could also enhance students’ perception of the FC approach. This idea is supported by results shown by Hampton and colleagues that indicate nursing students preferred sessions with audience response clickers [58]. This suggests that the inclusion of gamification methods or active participation can have positive outcomes for learning. Similarly, students who achieved better grades in a veterinary biochemistry and metabolism course had statistically significant higher participation in Poll Everywhere questions, another audience response system [59]. A review on the use of Kahoot! in anatomy, histology, and medical education classes has also shown predominantly positive effects on student outcomes, including improved collaborative learning, knowledge of content, attendance, and participation [60]. We ourselves have observed that the application of a FC-based system with elements of gamification and collaborative work in anatomy practical sessions can have positive effects [35,61].

Currently, a wide range of teaching tools are available, opening possibilities to create educational strategies that integrate these tools effectively to develop teaching models that meet each student’s unique needs, using the best methods from various instructional approaches. This would help provide a personalized education that matches each student’s learning style and preferred way of studying. The variation in students’ experiences and perceptions can be caused by numerous factors. Organizing focus groups with students to examine how they approach different types of teaching will likely shed light on each one’s learning methods and provide important data to organize teaching in the most effective way for each type of student

### 4.4. Limitations of the Study

In this study the evaluation of academic results was conducted collectively, encompassing both FC and traditional lecture-based classes, making it difficult to determine the specific impact of the FC approach on student performance. Considering this perspective, it would be intriguing to specifically compare the effects of the FC approach on final grades and the long-term retention of knowledge and skills in basic subjects within veterinary education.

Another limitation of the study lies in the restricted format of pre-class material (videos and annotated radiographs). Although our results indicate that students appreciated the provided materials, it would be interesting to assess their evaluation of other information supports, including traditional textbooks or more innovative resources that harness advanced technologies and game-like elements. On the other hand, a detailed study of the total time spent by students and its differentiation into pre-class and post-class time for each teaching method would provide interesting data on the effectiveness of both methods in student learning. Future research may benefit from addressing these aspects to provide a more comprehensive understanding of the FC effectiveness in veterinary basic subject learning.

## 5. Conclusions

Our study describes the implementation of the FC approach over two semesters in integrated classes covering anatomy and embryology, biochemistry, and physiology for first-year veterinary students. The results reveal that students actively engaged in both the preparation and execution of the in-classes sessions, and they highly appreciated the pre-class material provided. Feedback collected at the end of the first semester indicates that students held a favorable view of FC. Nevertheless, by the end of the second semester, a larger proportion of students showed a leaning toward traditional lecture-based classes over flipped ones. Interestingly, a substantial percentage of students reported no definitive preference between the two teaching methods, underscoring the potential importance of implementing blended learning strategies in the instruction of basic veterinary subjects.

The FC approach in veterinary medicine shows potential benefits in terms of student engagement, preparation, and satisfaction. However, careful consideration of material selection, coordination among instructors, and addressing workload concerns is essential for successful implementation.

## Figures and Tables

**Figure 1 vetsci-11-00354-f001:**
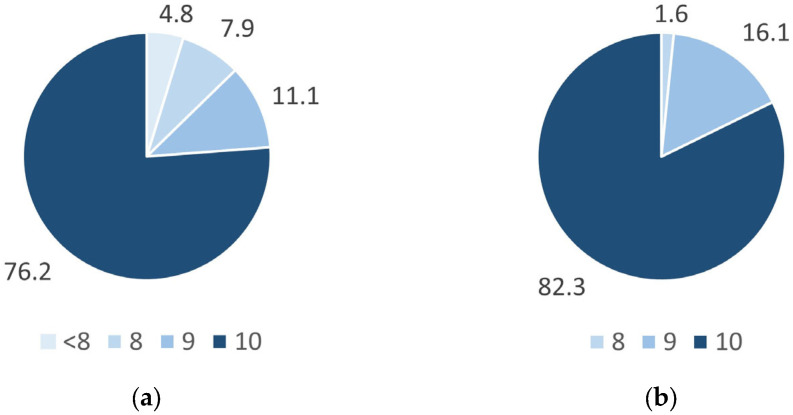
Distribution of the scores assigned by the students to the first video (**a**) and the second video (**b**) provided in physiology. The results are expressed as percentages, which are calculated by dividing the number of students who were assigned a score lower than 8 or equal to 8, 9, and 10 by the total number of students who viewed the corresponding video (63 for the first video and 62 for the second video).

**Figure 2 vetsci-11-00354-f002:**
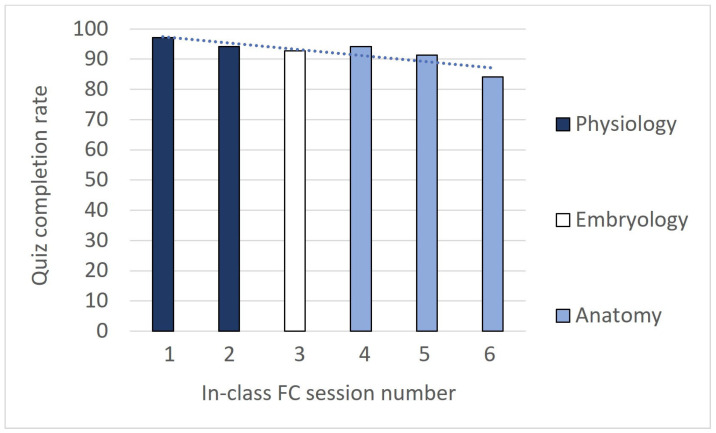
Percentage of students who completed the quiz on pre-material content for each SF1 in-class FC session. The results are shown in chronological order. A linear trend curve is shown in the dotted blue line. The calculation of its slope resulted in a value of −2.

**Figure 3 vetsci-11-00354-f003:**
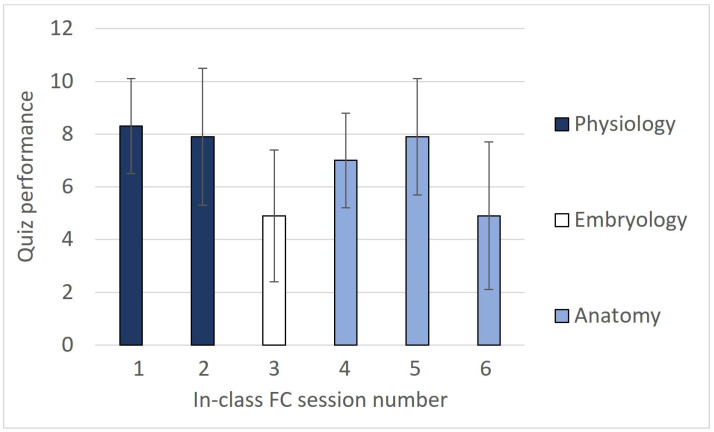
Means and standard deviations for the scores of the six quizzes in SF1, with the highest possible score being 10.

**Figure 4 vetsci-11-00354-f004:**
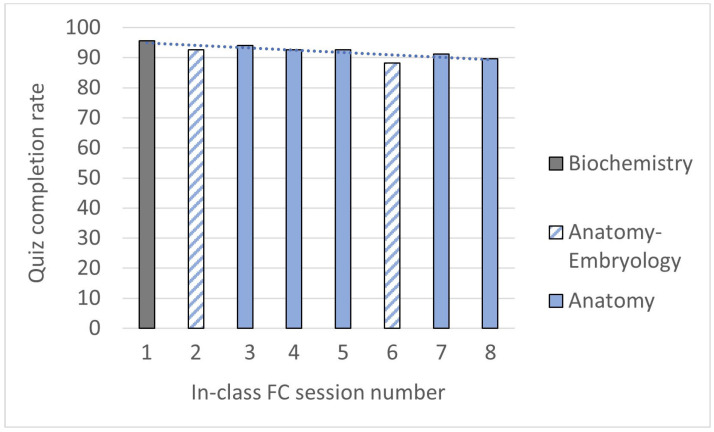
Percentage of students who completed the quiz for each SF2 in-class FC session. The results are shown in chronological order. A linear tendance curve is shown in the dotted blue line. The calculation of its slope resulted in a value of −0.8.

**Figure 5 vetsci-11-00354-f005:**
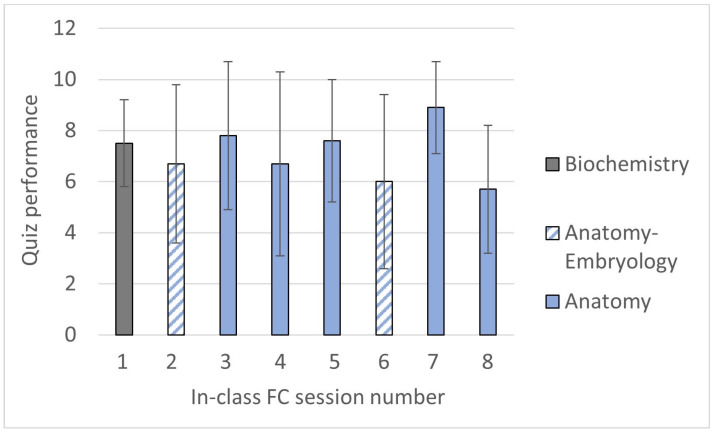
Means and standard deviations for the scores of the eight quizzes in SF2, with the highest possible score being 10.

**Figure 6 vetsci-11-00354-f006:**
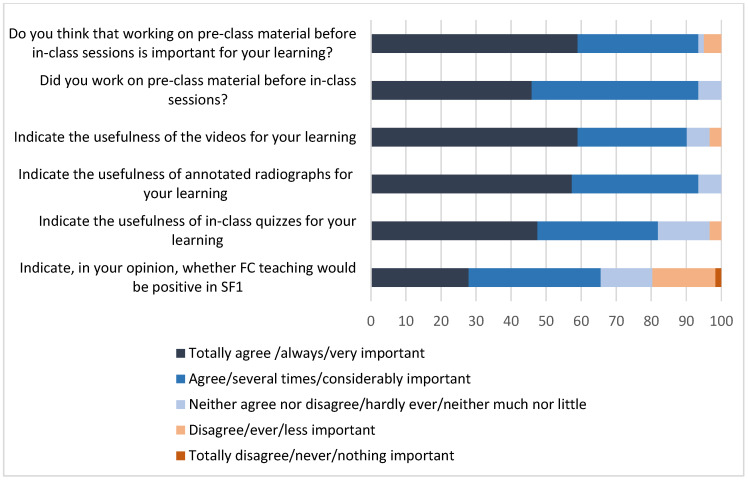
Results of the survey assessing student perception of FC. The survey was carried out at the end of the first semester, at the end of the SF1 subject. Results are shown as percentages.

**Figure 7 vetsci-11-00354-f007:**
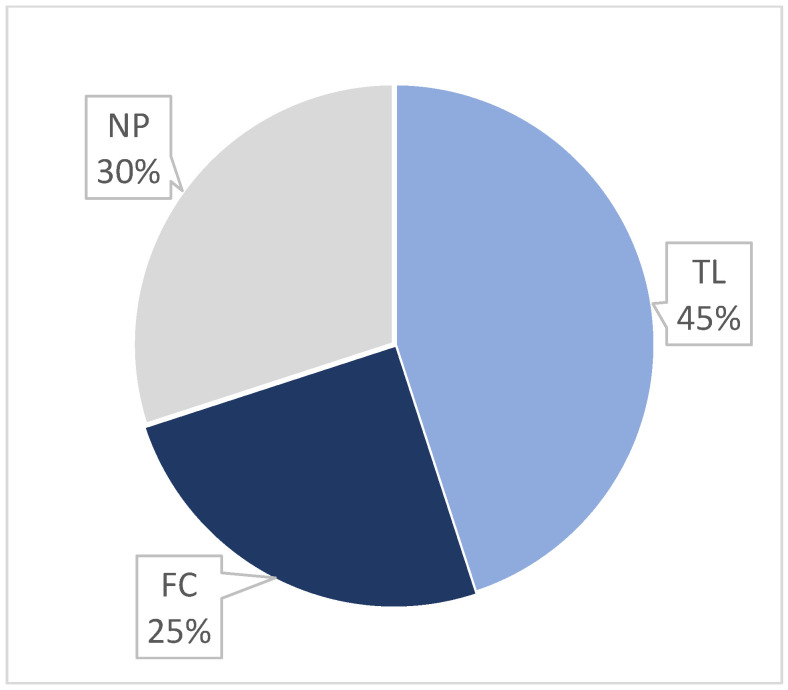
Distribution of students’ preferences regarding the teaching method. TL: Traditional Lectures; FC: Flipped Classroom; NP: No Preference.

**Table 1 vetsci-11-00354-t001:** FC topics. FC was implemented across various subjects in the first and second semesters.

	First Semester (SF1)		
In-Class FC Session	Subject	Topic	Duration (h)
1	Physiology	Introduction to cell biology	1
2		Cellular transport	1
3	Embryology	Embryonic manipulations	1
4	Anatomy	Introduction to locomotor apparatus	1
5		Radiology of the axial region	1
6		Radiology of limb pathologies	1
	**Second semester (SF2)**		
**In-class FC session**	**Subject**	**Topic**	**Duration (h)**
1	Biochemistry	Introduction to chemical reactions in cells andbiochemistry of solutions	2
2	Anatomy-Embryology	Pharyngeal region development	1
3		Nasal cavity and pharyngeal region Larynx and trachea	2
4	Anatomy	Lungs, diaphragm, and mediastinum	1
5		Radiology of the thoracic cavity	1
6	Anatomy-Embryology	Urinary apparatus	2
7	Anatomy	Abdominal cavity	1
8		Genital apparatus	2

**Table 2 vetsci-11-00354-t002:** Number of questions per quiz and score assigned to each question.

	Quizzes First Semester (SF1)		
In-class FC Session	Number of Questions	Points per Question	Total Points
1	8	1.25	10
2	4	2.5	10
3	4	2.5	10
4	4	2.5	10
5	5	2	10
6	10	1	10
	**Quizzes second semester (SF2)**		
**In-class FC session**	**Number of questions**	**Points per question**	**Total points**
1	10	1	10
2	5	2	10
3	6	1.67	10
4	5	2	10
5	5	2	10
6	10	1	10
7	5	2	10
8	10	1	10

## Data Availability

The data presented in this study are available upon request from the corresponding author.

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
