# Peer review of "Flipping Veterinary Biochemistry, Anatomy, and Physiology: Students’ Engagement and Perception"

_vetsci, 2024, doi:10.3390/vetsci11080354_

Round 1

Reviewer 1 Report

Comments and Suggestions for Authors

Table 1. Fix formatting to make it clear which subject the topics link to.  Either add borders or bring the subject to the top of the cells.

I would also include the number of hours each session is.  I was unable to determine if all sessions were the same length and how the length of the pre-work compared to the length of the session.

Line 189 states quizzes consisted of 4-10 questions but quiz performance (Figs 3 and 5) indicate the highest score is 10 per quiz.  Were these normalized?  How so?  How many questions were on each quiz?  This could help explain some of the differences in deviation.

Figure 1. The figure explaination doesn't seem to be what is shown in the graph.  Less than 8, 8, 9, and 10 are all calculated and graphed, but the legend suggests only two categories (8 or higher and less than 8).

Line 285 describes a relationship between the best fit lines, it may be useful for you to report the slopes.

Line 511 states that academic results were conducted collectively however only quiz performance for FC sessions are reported.  I would have liked to see quiz performance following TL.

Future directions could also include looking into post-class study time as FC aims to put the emphasis on pre-work instead while TL depends on greater post-work.

Comments on the Quality of English Language

Line 314, comments are presented below, not above.

Line 343, comments are presented below, not above.

Line 401, there is an extra "e" between "areas" and "such."

Author Response

Reviewer 1

First of all, we want to thank the Reviewer for constructive comments on our work.

Next, we will address his/her questions point by point.

Table 1. Fix formatting to make it clear which subject the topics link to.  Either add borders or bring the subject to the top of the cells. I would also include the number of hours each session is. I was unable to determine if all sessions were the same length and how the length of the pre-work compared to the length of the session.

Thank you for detecting the formatting problems of the table. We have separated the subjects with borders in order to better differentiate the topics and we have included a new column in the table with the hours dedicated to each in-class FC session.

Line 189 states quizzes consisted of 4-10 questions but quiz performance (Figs 3 and 5) indicate the highest score is 10 per quiz.  Were these normalized?  How so?  How many questions were on each quiz?  This could help explain some of the differences in deviation.

We have included a new table (Table 2) indicating the number of questions in each quiz. Indeed, the score for each quiz was normalized to 10 points by assigning each question the appropriate value (as indicated in the table) to normalize to 10.

An explanation of this has been added at that part in the Materials and Methods section.

Figure 1. The figure explanation doesn't seem to be what is shown in the graph.  Less than 8, 8, 9, and 10 are all calculated and graphed, but the legend suggests only two categories (8 or higher and less than 8).

We have rewritten the sentence as follows:

“The results are expressed as percentages, which are calculated by dividing the number of students who assigned a score lower than 8 or equal to 8, 9 and 10., by the total number of students who viewed the corresponding video”.

Additionally, we have changed part b) of that figure, as we realized there was an error (part a of the figure was duplicated) which, in any case, does not alter the message.

Line 285 describes a relationship between the best fit lines, it may be useful for you to report the slopes.

We have calculated the slopes, which results in -2 for the first semester (figure 2) and -0.8 for the second semester (figure 4). We have included these data on the figure legends and in the text (line 295).

Line 511 states that academic results were conducted collectively however only quiz performance for FC sessions are reported.  I would have liked to see quiz performance following TL.

The quizzes were only administered in the FC classes, and therefore, unfortunately, it is not possible to compare both types of teaching in this aspect. On the other hand, the course organization makes it impossible to differentiate the assessment of the two systems, as the exams include questions on topics explained either through FC or TL. This is why we consider this aspect a limitation of our study.

Future directions could also include looking into post-class study time as FC aims to put the emphasis on pre-work instead while TL depends on greater post-work.

We thank the reviewer for their suggestion, and in this regard, we have included the following phrase at the end of section 4.4.

“On the other hand, a detailed study of the total time spent by students and its differentiation into pre-class and post-class time for each teaching method would provide interesting data on the effectiveness of both methods in student learning.”

Finally, we thank the reviewer for detecting the typographical and English errors on the lines 314, 343 and 401 of the original manuscript; we have corrected those mistakes.

Reviewer 2 Report

Comments and Suggestions for Authors

This is a clear, relevant, easy to follow study on implementation of flipped classroom learning activities in preclinical veterinary science. 

The authors can increase the usefulness and relevance of the study and its findings by the following:

2.2 Describe the relevant demographics of the participating students, specifically recent/non recent school leavers, numbers learning in a second language, with neurodiversity or learning disabillities if information is available. This is to help the reader assess how representative this group of students are.

2.3.1 Describe in more detail the nature of the videos. How were the 15-20 min  subtopics for each segment identified/grouped? Were they primarily existing lectures and powerpoint slides recorded for students to watch at home? Where they produced with professional assistance for production?  how were questions included in the videos- did students need to complete them to proceed? Were these materials delivered on a learning management system and if so were student views/time spent watching them recorded and analysed?  

2.3.2. How was the workload for students determined? Did students confirm it was sufficient? Did the researchers follow methods for workload measurement, what recording tools were used, or were  there other measures such as time students took to complete online activities that were analysed? 

2.3.4 What preparation and support did students receive to assist them in adapting to the flipped classroom learning activities?

Is it correct to conclude that the in class sessions occurred in the same learning spaces as lectures- e.g. tiered lecture theatre, or another space?

How was the feedback provided on the quizes- other than class discussion, was there also online feedback to all quizes?   It is not clear if students could complete the quizes online at other times, or only in class.

The authors must describe the nature of the assessment of SF1 and 2- were the assessments constructively aligned to the learning approach that students were required to take?  What information did students receive about assessment and how to prepare for it?

Results

REport on workload for students- the amount that the researchers estimated and the amount that students actually undertook.  

Figure 3 - Why were the results of quiz 3 and 6 fairly poor relative to the remainder of the topics?  How were the questions checked for validity, reliability and appropriate level of difficulty?

Describe the feedback on learning that students received, from the quizes as it is not clear if this was just the in class discussion.

Discussion. 4.1 and 4.2 The introduction and discussion both point out that flipped classroom approaches to teaching and learning can be very varied, and range from effective and well received, to much less useful, depending on aspects of design and implementation.  Comment further in the discussion on  the elements that were shown to be most effective and engaging in this study, including workload management, and compare to the literature. 

Limitations. Comment on the variation in experience and perceptions that is likely to exist in the student group and research methodologies that could be used to better understand the reason for this variation.  

Author Response

Reviewer2

We would like to thank Reviewer 2 for their comments and suggestions, which we believe have helped improve the text of the paper. We have addressed all of them as shown below.

2.2 Describe the relevant demographics of the participating students, specifically recent/non recent school leavers, numbers learning in a second language, with neurodiversity or learning disabillities if information is available. This is to help the reader assess how representative this group of students are.

As indicated in section 2.4 Surveys and data analysis (line 229), no demographic data were collected for this study. Unfortunately, the working conditions did not allow for this possibility, which would undoubtedly have enabled a more in-depth analysis of the data.

In any case, regarding the factors mentioned by the reviewer, we can confirm that language was not a determining factor since all the students had French as their first language, the language in which they received instruction. Additionally, there were no reported cases of learning disabilities.

2.3.1 Describe in more detail the nature of the videos. How were the 15-20 min subtopics for each segment identified/grouped? Were they primarily existing lectures and powerpoint slides recorded for students to watch at home? Where they produced with professional assistance for production?  how were questions included in the videos- did students need to complete them to proceed? Were these materials delivered on a learning management system and if so were student views/time spent watching them recorded and analysed?  

The videos were recorded by the instructors (teachers of the subjects) who had previous training in recording and editing audiovisual material. For their production and duration, the coherence of the content included in the videos was taken into account.

In some videos, questions were posed either orally or in writing. The answers were provided throughout the video and students did not need to formally respond to these questions to proceed. We have removed the word "interactivity" from the text because we thought it might lead to a misinterpretation.

On the other hand, the material management system did not allow us to see the time students spent on each video. This is now indicated at the end of the paragraph.

We have reviewed the explanation and modified it as follows, with the intention of clarifying these aspects (2.3.1.):

“The teaching videos were created by the instructors (academic teachers trained in video recording and editing), integrating a coherent block of content into each one. The videos were created using PowerPoint slides that were adapted to provide appropriate support for educational videos. When necessary, the videos were further edited using Adobe Premiere Pro. The videos included oral explanations given by the professors, diagrams, photos, animations, and short clips. Key words and sentences were also integrated to assist students in grasping the central concepts. The content was organized with a clear structure of chapters, and subchapters when needed. Questions were eventually included as a teaching aid to promote student attention and engagement during viewing. Each video was conceived as an independent document and could be viewed independently of the others. Student views or the time spent watching these materials were not recorded.”

2.3.2. How was the workload for students determined? Did students confirm it was sufficient? Did the researchers follow methods for workload measurement, what recording tools were used, or were  there other measures such as time students took to complete online activities that were analysed? 

2.3.4 What preparation and support did students receive to assist them in adapting to the flipped classroom learning activities?

The students' workload was calculated subjectively and approximately by the teachers, based on previous experience (over 10 years of experience teaching the subjects) and the length of the documents provided, so that the estimated preparation time did not exceed 1 hour for each hour of class work.

On the other hand, at the beginning of the FC sessions, the students were instructed on the working method, insisting on the fact that previewing the videos and studying them before the classes was fundamental for later work in the classroom sessions, which would not be developed as lecture explanations, but as sessions for doubts and in-depth study of certain aspects of the syllabus.

We have modified the explanation in the text as follows:

“For the preparation of each in-class FC session, students received oral instructions at least once in the classroom, along with a single e-mail. At the beginning of the FC sessions, the students were instructed on the working method, insisting on the fact that previewing the videos and studying them before the classes was fundamental for later work in the classroom sessions, which would not be developed as traditional lecture explanations, but as sessions for doubts and in-depth study of certain aspects of the syllabus. To facilitate self-organization, students were informed of the estimated amount of personal working time required to prepare each FC session. This estimation was calculated subjectively by the instructors based on previous experience, the duration of the video(s) or the number of radiographs, as well as the complexity of the content.”

Is it correct to conclude that the in class sessions occurred in the same learning spaces as lectures- e.g. tiered lecture theatre, or another space?

Yes, that is correct; the classes took place in the same spaces for both types of teaching.

We have included a sentence indicating this in section 2.3.1.:

“Both the FC and the traditional learning were conducted in the same classrooms.”

How was the feedback provided on the quizes- other than class discussion, was there also online feedback to all quizes?   It is not clear if students could complete the quizes online at other times, or only in class.

The first time students saw and completed the quizzes was in class. Afterwards, the quizzes remained available for review, before final assessments.

After completing the quizzes, the students received online feedback through the online correction system that allows them to identify which answers they got right and which ones they got wrong.

We will address deeper this question later, as there is another question on the same topic.

The authors must describe the nature of the assessment of SF1 and 2- were the assessments constructively aligned to the learning approach that students were required to take?  What information did students receive about assessment and how to prepare for it?

We have included this explanation at the 2.3.5. section.

“The theoretical evaluation (which affects the teaching described in this work) of SF1 and SF2 is carried out through theoretical exams featuring multiple-choice questions and short-answer questions. Students are informed about the types of exams from the beginning of each semester. The exams combine questions that directly assess the knowledge acquired by the students with others that evaluate deeper learning, connecting concepts. To help students familiarize themselves with the exam format, various practice sessions and different types of tests were conducted, including in-class Kahoot quizzes and self-assessment tests outside the classroom.”

Results

Report on workload for students- the amount that the researchers estimated and the amount that students actually undertook.  

As mentioned earlier, the instructors subjectively calculated the workload based on their prior experience. Measuring the actual study time of students outside the classroom is challenging, as previous studies indicate significant variation in the time students report dedicating to out-of-class study. We address this issue in a later paragraph.

Figure 3 - Why were the results of quiz 3 and 6 fairly poor relative to the remainder of the topics?  How were the questions checked for validity, reliability and appropriate level of difficulty?

We thank the reviewer for this observation. The review of the quizzes with lower performance shows that both had questions with a lower percentage of correct answers. Analysis of these questions revealed that they addressed issues that were not directly explicated in the pre-class material and, therefore, required a higher degree of reflection or knowledge. We have included a paragraph describing this condition at section 3.1.1. as follows:

“The analysis of the responses showed that the questions with the lowest percentage of correct answers in those quizzes were those whose answers did not come explicitly from the provided material content but required a higher degree of reflection or interpretation. This, apparently, presented a greater level of difficulty for the students compared to the other quizzes.”

Describe the feedback on learning that students received from the quizes as it is not clear if this was just the in class discussion.

We have included a sentence at the endo of 2.3.4. section explaining that:

“The students received immediate feedback on their quiz results through the online correction system. Additionally, the proposed questions were discussed in class, with particular focus on those where the results had been weaker.”

Discussion. 4.1 and 4.2 The introduction and discussion both point out that flipped classroom approaches to teaching and learning can be very varied, and range from effective and well received, to much less useful, depending on aspects of design and implementation.  Comment further in the discussion on the elements that were shown to be most effective and engaging in this study, including workload management, and compare to the literature.

In order to highlight the most effective elements of this study, we have introduced a sentence at the beginning of the discussion (lines 414-415) indicating:

“In this study, the most effective elements for FC learning, according to the students, are the materials provided before class. While the evaluation quizzes used at the beginning of the face-to-face sessions were well received, they did not reach the same level of acceptance. Additionally, one of the factors that may influence the acceptance of the FC is that some students consider this type of teaching to be more time-consuming.”

Furthermore, we have introduced a paragraph to discuss the fact that some students consider the flipped classroom to be more time-consuming. This has been added at line 525 indicating:

“An important aspect to consider in the development of FC is that this type of teaching, according to students, sometimes requires more dedication time, mainly because the traditional lecture format requires less out-of-class preparation [33,52,53]. The previewing of videos, supplementary work materials, or group discussions that FC often involves can be perceived by students as more time-consuming [54,55]. In our case, although instructors took special care to ensure that out-of-class work with provided materials did not result in an excessive increase in dedication time, some students did not perceive it this way in the second semester. This was likely because the number of FC sessions increased compared to the first semester. In this context, combining different teaching approaches may improve the overall educational experience for students.”

Limitations. Comment on the variation in experience and perceptions that is likely to exist in the student group and research methodologies that could be used to better understand the reason for this variation. 

We have included a brief paragraph at the end of 4.3. to note these considerations:

“The variation in students' experiences and perceptions can be caused by numerous factors. Organizing focus groups with students to examine into how they approach different types of teaching will likely shed light on each one's learning methods and provide important data to organize teaching in the most effective way for each type of student.”

Reviewer 3 Report

Comments and Suggestions for Authors

It is my understanding the this article  is highly relevant as it discusses the implementation of the flipped classroom (FC) approach in veterinary education. It addresses a significant gap in educational methodologies within veterinary studies, where traditional teaching methods have predominated. By exploring the FC model's application in veterinary education, the study contributes to a growing body of literature advocating for modern teaching methods in health sciences. It encourages further exploration of blended learning approaches that combine traditional and innovative methods. The text also addresses the evolving preferences of students over time, with a notable shift towards traditional lectures in the second semester. This aspect is critical for educators to consider when implementing FC, as it underscores the need for a balanced approach that mitigates workload concerns. Overall, the text is relevant as it not only contributes to the academic discourse on veterinary education but also provides practical insights for educators aiming to enhance learning experiences through innovative teaching strategies

Author Response

We would like to thank Reviewer 3 for his/her positive feedback on our work.